

# Squirrel monkey responses to information from social demonstration and individual exploration using touchscreen and object choice tasks

Elizabeth Renner, Mark Atkinson and Christine A. Caldwell

Psychology, Faculty of Natural Sciences, University of Stirling, Stirling, UK

## ABSTRACT

We aimed to study whether a non-human primate species responded differently to information acquired socially compared with that acquired individually. To do so, we attempted to train squirrel monkeys to perform binary discriminations. These involved exposure to either social information (human or puppet demonstrator performs an initial 'information trial') or individual exploration (monkey performs information trial as well as subsequent test trials). In Experiment 1, we presented the task on a touchscreen tablet. Only one monkey appeared to learn the significance of the information trial, and across the group there was no improvement in performance over sessions. The proficient individual showed little evidence of successful transfer to three-way discrimination problems, suggesting limited representation of the task structure. In Experiment 2, we used a logically identical task, presented as a physical object choice (inverted cups concealing a food reward). No monkeys learned to use the information trial cues, and success again did not increase over sessions. We concluded that the monkeys' poor performance in Experiment 1 was not attributable to the mode of presentation (touchscreen), but reflected real difficulties with mastering the task structure. For both experiments, we analysed the monkeys' spontaneous responses to the different trial types (social-win, social-lose, individual-win, and individual-lose). We found that monkeys had a tendency to repeat selections made during the information trial, whether these were made by themselves or by a demonstrator. This tendency to repeat was observed even following lose trials (i.e. when incorrect). Apparent 'success' following win trials was probably largely an artefact of behavioural inertia (individual learning conditions) and stimulus enhancement (social learning conditions), rather than sensitivity to the reward cues associated with that stimulus. Although monkeys did respond somewhat differently (more repeats) following win trials, compared with lose trials, this was no more apparent in the object choice task than the touchscreen task, again suggesting that the less ecologically valid presentation medium did not actively disrupt potential for learning the discrimination rule. Both touchscreen and physical object choice tasks appear to be valid methods to study learning in squirrel monkeys, with neither method giving a clear performance advantage over the other. However, this population did not master the contingencies in these tasks.

Corresponding author
Elizabeth Renner,
elizabeth.renner@stir.ac.uk

## INTRODUCTION

Some explanations of the differences between human culture and the behavioural traditions of other species have suggested that humans are particularly attentive to social information, or even that humans possess specialised cognitive mechanisms for acquiring information from social sources (*Tomasello, Kruger & Ratner, 1993*; *Tennie, Call & Tomasello, 2009*; *Csibra & Gergely, 2011*; *Dean et al., 2012*). However, it remains difficult to directly compare the effects of exposure to information from a social source with information obtained from one's own experience, since these typically cannot be regarded as equivalent.

Whenever information is potentially generalisable, the inference drawn from observation of another's activity is liable to be very different to that drawn from one's own naïve exploration, even when the action itself is identical. For example, consider being faced with a combination lock on a door, for which you do not know the code. Tapping in a random code yourself (unsuccessfully) will likely lead you to conclude that the code is incorrect. However, observing someone else tapping in a code unsuccessfully might lead you to an entirely different conclusion, or at least to entertain a number of alternative possibilities, many of which would likely entail the information contained within the other's actions having some potential value, despite the unsuccessful outcome. For example, you might conclude that the actor knew the code, but entered one or more digits incorrectly. Alternatively, the code could have been entered correctly but the mechanism was broken, or the actor simply failed to use sufficient force in turning the lock.

The non-equivalence of information acquired from observation of others, compared with personal experience, arises as a consequence of differential knowledge of the prior behavioural history of the actors in question (complete for ourselves, but usually very limited for others). As a result, in many situations learners would be expected to respond differently to information acquired socially compared with that acquired from individual exploration. Such differential responding would be predicted from either reasoning-based or associative learning accounts, even if the fundamental mechanisms underlying that learning were assumed to be the same.

Theories that propose specialised mechanisms for processing social information in humans would likely predict that humans and non-human primates (NHPs) treat social information somehow differently (*Tomasello, Kruger & Ratner, 1993*; *Tennie, Call & Tomasello, 2009*; *Csibra & Gergely, 2011*; *Dean et al., 2012*). However, it is unclear how this difference might manifest in terms of relative responsiveness to information obtained from a social source vs information acquired through one's own personal experience. One possibility is that humans would be more likely (compared with NHPs) to repeat behaviours they had seen others perform, even when the behaviours had not appeared to produce any reward, in a manner that would not occur under individual learning conditions (i.e. following their own performance of an unreinforced response). The tendency of humans to 'overimitate' others' irrelevant actions in causal sequences has been
well documented (*Horner & Whiten, 2005*; *Lyons, Young & Keil, 2007*; *Whiten et al., 2016*). However, we do not yet know the extent to which humans or NHPs 'overimitate' their own unreinforced actions in individual learning conditions; therefore, this first possibility is yet to be empirically assessed. Alternatively, human use of social information might be relatively proficient compared with use of information from direct personal experience, for example, showing little reduction in learning fidelity, or possibly even equivalent performance, in relation to the likelihood of repetition of reinforced, over unreinforced, responses. This would be expected to be relative to any such pattern observed in the social and individual learning of NHPs, for whom reduced sensitivity would be predicted for vicarious feedback compared with directly experienced reinforcement contingencies.

Thus, it is problematic to evaluate the claim of specialised mechanisms for processing social information in humans based on existing knowledge of the behaviour of either humans or their closest extant evolutionary relatives. This research aims to contribute to the knowledge base upon which such claims could be evaluated. In the studies reported here we attempted to train squirrel monkeys (*Saimiri sciureus*), a platyrrhine monkey, to learn a discrimination rule using information from either a social demonstration or direct personal experience, with a view to better understanding how these primates respond to information depending on source. In this task the information provided is purely episodic, relating only to that particular problem, and with perfect predictive value. And since the information value of 'successful' and 'unsuccessful' trials is exactly equivalent across both the social and individual learning conditions, then in principle the performance of any subjects achieving mastery of the task contingencies can be used to evaluate whether source alone affects responses to (otherwise directly comparable) information. We aimed to train monkeys to a pre-set performance criterion which would indicate that they had learned the predictive value of the information available (whether from a social demonstration or their own experience) and could use this to find a rewarded stimulus on subsequent trials involving the same set of stimuli. As well as allowing us to compare the response patterns of those exposed to social information vs personal experience for individuals who reach criterion (i.e. episodic information use), this would also allow us to compare rates of learning between monkeys assigned to a social learning condition and those assigned to an individual learning condition (i.e. achievement of generalisable task competence).

In this task, information trials could consist of selection of either the rewarded or the unrewarded stimulus. Because the task involves binary discriminations, the information received about a response indicates that it should be either repeated (rewarded stimulus selected, or 'win' trial) or avoided (unrewarded stimulus selected, or 'lose' trial). Successful performance therefore would encompass both reliable re-selection of rewarded stimuli and reliable avoidance of unrewarded stimuli. Within the context of a binary discrimination task, the two information trial types (rewarded and unrewarded) both provide unambiguous information about the location of the reward, and are therefore (in principle) equally informative, assuming the discrimination rule has been learned. We hoped that this format would allow us to examine whether error rates were different following rewarded and unrewarded information trials, and if so, whether this pattern

differed between subjects exposed to social information and those learning from their own experience.

Discrimination learning tasks such as the one we use here have been widely employed in studies of animal learning and developmental psychology. These studies have provided evidence of the formation of 'learning sets' in children and NHPs (*Harlow, 1949*; *Levinson & Reese, 1967*). However, most have involved learning either only from personal experience, with information trial and test trial both completed by the participant (*Berman, Rane & Bahow, 1970*; *Berman, 1971*, with children), or only from social information, with vicarious observation of the information trial (*Templeton, 1998*, with starlings; n.b., we use 'vicarious' throughout to refer to any kind of indirect exposure occurring as a consequence of events that are neither generated, nor directly experienced, by the learner). We know of only one case where similar paradigms have been used to compare performance following exposure to social information with that following individual exploration (*Monfardini et al., 2012*, with macaques and adult humans). The research questions of *Monfardini et al. (2012)* were rather different from our own, examining how well information was used under significant memory load, with information and test trials separated by several intervening trials. Nonetheless, they found that rewarded and unrewarded trials had differing effects on performance depending on whether they were performed by the subjects themselves or by another monkey. In the individual learning condition, monkeys performed well following a rewarded trial and poorly following an unrewarded trial; but in the social learning condition, the outcome of the information trial did not differentially affect performance.

We first report our attempt to train squirrel monkeys on a touchscreen version of the task (Experiment 1). Touchscreen tasks have proven useful in making cognition questions feasible to explore in zoo settings (*Egelkamp & Ross, 2019*), as touching a screen is a fairly straightforward behaviour to train some animals to do. Such tasks have the additional benefit of allowing researchers to escape the 'one-off' nature of object-based tasks (*Renner, 2015*) and expose an individual to multiple trials, which minimises the effects of outlier days or events and enables repeated measurement of a phenomenon. Squirrel monkeys have been shown to master use of a touchscreen in an experimental setting (*Kangas & Bergman, 2012*), though our specific population of monkeys had limited experience of interaction with touchscreens. In Experiment 1, the squirrel monkeys' success rates were very low, and showed little evidence of improvement over time. To examine whether this was due to the presentation format, we also attempted to train this population using a three-dimensional object-choice version of the task (Experiment 2). The limited success of the squirrel monkeys (on both tasks) meant that we were unable to fully address our original aims of comparing the relative efficacy of social and individual learning and comparing error rates for information trial types. Nevertheless, here we document the monkeys' performance on both tasks. This includes comparisons of the monkeys' performance depending on whether they received information from social demonstrations or their own experience, and of how responses differed following rewarded vs unrewarded information trials. We also compare performance across the two presentation mediums.

# EXPERIMENT 1: TOUCHSCREEN

## Materials and methods

### Participants

Approval for this project was provided by the Living Links to Human Evolution Research Centre at RZSS Edinburgh Zoo. At the commencement of this research, a total of 30 squirrel monkeys were housed at the Living Links to Human Evolution Research Centre at Edinburgh Zoo, in two social groups, East and West (*MacDonald & Whiten, 2011*). Monkeys had previously been trained with positive reinforcement to enter the research area and allow themselves to be temporarily separated (for a maximum of 15 min) from their social group. Research participation was voluntary. The research area for each group consisted of eight cubicles that could be separated from one another via the use of sliding doors. When separated, monkeys were allowed to rejoin their social group after indicating a desire to do so by making manual contact with the sliding door. Characteristics of monkeys that participated in this study are presented in Table 1. A total of 13 monkeys (eight from the East group and five from the West group) passed the touchscreen-training phase of this study (described below). A further eight monkeys participated in touchscreen training but did not pass the training phase, and nine monkeys never participated in touchscreen training.

### Ethics statement

Ethical approval was granted by the University of Stirling Psychology Ethics Committee. Animals were not food or water deprived. The research was conducted in line with the guidance provided by *The Association for the Study of Animal Behaviour (2015)* in *Guidelines for the Treatment of Animals in Behavioural Research and Teaching,* and *The British Psychological Society (2012)* in *Guidelines for Psychologists Working with Animals.*

### Apparatus

A Microsoft Surface 4 touch-sensitive capacitive tablet computer was used to present the training and experimental tasks to the monkeys. The tablet was attached to a rolling projector stand via a metal frame and a mobile arm (Ergotron, Minneapolis, MN, USA). The custom programme for displaying the task was written in PsychoPy (*Peirce et al., 2019*). A digital video camera was used to record video of experimental sessions.

### Training procedures

Prior to beginning the experiment, monkeys were trained via shaping (*Pryor, 2002*) to make manual contact with the tablet or the metal frame bordering the touch-sensitive surface of the tablet. Once they reliably did so, they were trained to touch the active area of the screen; next, to touch a large stimulus (~11 cm in diameter) presented on the screen; and finally, to touch a smaller stimulus (~5.5 cm in diameter) presented on the screen. When monkeys correctly touched the screen or a stimulus at each respective training level, a visual reward cue appeared, an auditory click sound was played by the tablet, and the monkey was rewarded by the experimenter with a raisin. The visual reward cue for correctly touching training stimuli was a 'sunburst' image (see Fig. 1B).
**Table 1  Participant characteristics.**

| Name | Sex | Group | Age (as of Feb. 2016) | Condition | Participation in Expt 1 | No. sessions, Expt 1 | Participation in Expt 2 | No. sessions, Expt 2 |
|------|-----|-------|------------------------|-----------|--------------------------|----------------------|--------------------------|----------------------|
| Amarilla | F | East | 4 | Individual | Training, testing | A: 44 B: 19 | Testing (insufficient sessions) | 9 |
| Boa | M | East | 10 | N/A | Training | 0 | N/A | 0 |
| Cali | F | East | 10 | Social | Training, testing | 42 | Testing | 20 |
| Ciara | F | East | 4 | Social | Training, testing | 55 | Testing | 19 |
| Dora | F | East | 5 | Individual | Training, testing | 69 | Testing | 20 |
| Elie | F | East | 10 | Individual | Training | 0 | Testing | 15 |
| Flora | F | East | 7 | Individual | Training, testing | 34 | Testing (insufficient sessions) | 3 |
| Gabriela | F | East | 4 | Social | Training, testing | 61 | Testing | 20 |
| Gisele | F | West | 3 | Social | Training, testing | 27 | Testing (insufficient sessions) | 0 |
| Jasmin | F | West | 13 | Individual | Training, testing | 14 | Testing | 14 |
| Lexi | F | East | 5 | Social | Training | 0 | Testing (insufficient sessions) | 4 |
| Loki | F | West | 1 | Social | N/A | 0 | Testing | 15 |
| Maya | F | East | 12 | N/A | Training | 0 | N/A | 0 |
| Orla | F | West | 4 | Individual | Training, testing | 52 | Testing (insufficient sessions) | 0 |
| Pelusa | F | East | 4 | Individual | Training, testing | 60 | Testing | 20 |
| Roca | F | East | 13 | Individual | Training | 0 | Testing (insufficient sessions) | 9 |
| Sancha | F | West | 6 | Social | Training, testing | 67 | Testing | 20 |
| Sipi | F | East | 6 | Social | Training | 0 | Testing (insufficient sessions) | 3 |
| Tatu | F | East | 14 | Social | Training | 0 | Testing (insufficient sessions) | 1 |
| Toomi | F | West | 9 | Individual | Training, testing | 64 | Testing (side bias) | 12 |
| Valencia | F | East | 4 | Social | Training, testing | 59 | Testing | 16 |
| Yendi | F | East | 7 | N/A | Training | 0 | N/A | 0 |

Notes:
For Experiment 1, completion of at least one T2 was required for the data from a session to be used. For Experiment 2, completion of at least one full problem (T1–T5) was required for the data from a session to be used. For Experiment 2, the data from monkeys who completed fewer than 10 sessions, indicated by '(insufficient sessions)', were not included in the analysis.
Expt, Experiment; F, female; M, male.

At each of the training levels in which monkeys touched stimuli, they were required to make contact with the relevant part of the screen on 8 or more out of the 10 trials presented in a session and within 30 s of the beginning of each trial, and to do this for three consecutive research sessions in order to move on to the next level. To progress to the testing phase, monkeys completed a minimum of eight training sessions with the touchscreen. One monkey progressed to the testing phase after eight training sessions; ten progressed to testing after nine training sessions; two progressed to testing after ten training sessions; and all others engaged in touchscreen training fewer than nine times and therefore never completed training.

### Testing procedures

Once monkeys passed the final touchscreen-training level, they were presented with the experimental task, which consisted of two stages, A and B. Task stimuli consisted of a

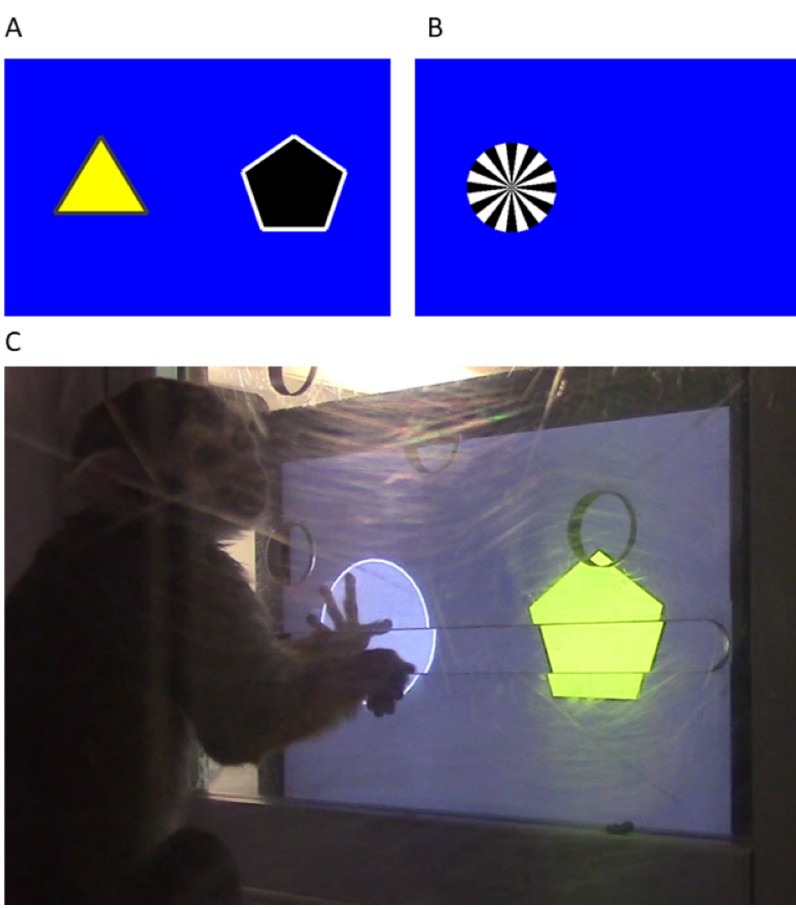

**Figure 1 Touchscreen example problem and apparatus setup.** (A) A sample two-stimulus (Stage A) problem. (B) Appearance of the screen (example) after the rewarded stimulus was selected, with the reward cue (sunburst) appearing at the location of the rewarded stimulus. (C) A squirrel monkey interacting with the touchscreen task.

variety of shapes (circles, triangles, diamonds, pentagons, and hexagons) of various colours (blue, yellow, black, white, and grey), taking into consideration the limited range of colours platyrrhine monkeys are likely to be able to discriminate (*Waitt & Buchanan-Smith, 2006*). Stimulus fill colour, outline colour, and number of sides, as well as background colour, were randomly generated by the task programme. For all trials in Stage A, paired stimuli (one on the left, one on the right) were presented; for all trials in Stage B, three stimuli were presented in a horizontal row on the screen. See Fig. 1 for an example.

Stage A. Experimental sessions consisted of a sequence of four problems. Each problem was presented for five trials: a single initial 'information trial' (T1) and four subsequent test trials (T2–T5) (see Table S1 for details). Although the first test trial (T2), which immediately followed the information trial, was the critical trial from the perspective of the analysis, additional trials were included for every problem as a means to reinforce the monkeys' learning of the overall task structure. Specifically, this was intended to help the monkeys learn that for any given pair of stimuli, only one was rewarded (and the other unrewarded), and that the identity and location of the rewarded stimulus always remained

the same within a problem, such that feedback from a selection made on one trial had perfect predictive value regarding reward status for all subsequent trials involving the same pair of stimuli. Stimuli did not change position between trials within a problem.

Initial selections were randomly assigned to be either rewarded or unrewarded. That is, we programmed the task to generate a list such as [U R U R], meaning that the stimulus selected on the information trial of the first problem was unrewarded (regardless of which position on the screen the stimulus occupied), and that the alternative stimulus would then be the rewarded stimulus. The stimulus selected on the information trial of the second problem was then the rewarded stimulus (regardless of its position), and so on. We programmed the task to ensure that half of information trials were rewarded and half were unrewarded. This was to help prevent monkeys from developing side biases. For example, if a monkey always selected the stimulus on the left in information trials, the left-side stimulus would be the rewarded stimulus only half of the time.

When a rewarded stimulus was selected, both stimuli disappeared, a sunburst visual reward cue appeared where the rewarded stimulus had been, and an auditory click was generated by the computer. Additionally, when a monkey selected a rewarded stimulus on any trial, they received a raisin. When an unrewarded stimulus was selected, all stimuli disappeared for a time-out of 3 s, no auditory cue was emitted, and no raisin was dispensed. The same stimulus, in the same location on the screen, was the rewarded stimulus for an entire problem (i.e. one information trial followed by four test trials using the same pair of stimuli).

Half of the monkeys were assigned to the individual condition, and the other half were assigned to the social condition. In the individual condition, at the beginning of a session, the tablet was placed within a monkey's reaching distance from the cubicle by use of a mobile trolley. Then monkeys themselves selected one of the stimuli in the information trial, and were rewarded (or not) accordingly. They were then allowed to select one of the stimuli on each of the four subsequent test trials. Between problems, the tablet was moved outside of reaching distance for several seconds before again being placed within reaching distance for the next problem.

In the social condition, at the beginning of a session, the tablet was placed outside of a monkey's reaching distance from the cubicle. Monkeys saw a social demonstrator select one of the stimuli in the information trial. The tablet was then placed in reaching distance to allow the monkey to perform the next four test trials. After this, the tablet was moved outside of reaching distance to prepare for the start of the next problem.

In half of the research sessions, the social demonstrator was the human experimenter, and in the other half, this was a puppet (a plush squirrel monkey toy with some of the stuffing removed; Wild Republic, Twinsburg, OH, USA) operated by the experimenter (Fig. 2). Researchers have successfully used a puppet demonstrator in a study with children (*Wood, Kendal & Flynn, 2013*), a stuffed conspecific demonstrator in a study with birds (*Truskanov & Lotem, 2017*), and videos of a human dressed in an ape-like costume in a study with great apes (*Krupenye et al., 2016*). Because of the previously successful implementation of these methods, we explored whether a puppet demonstrator would induce better performance than a traditional human demonstrator on this task. Previous

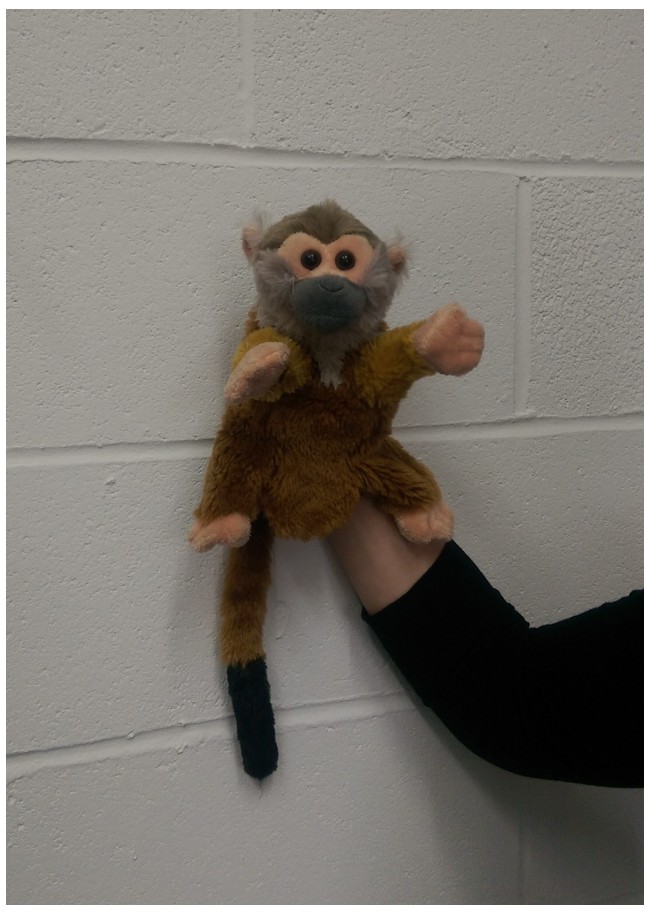

**Figure 2 The squirrel monkey puppet used as a demonstrator in Experiment 1.**

research also found that a 'monkey-like' human demonstrator, who did not solicit monkeys' attention and consumed rewards after rewarded trials, resulted in better subsequent performance than a traditional human demonstrator who solicited monkeys' attention and did not consume rewards (*Monfardini, Hadj-Bouziane & Meunier, 2014*). However, due to the potential for distraction, neither the experimenter nor the puppet received a food reward after selecting the rewarded stimulus on T1.

We were initially interested in whether monkeys would successfully transfer from Stage A (two stimuli) to Stage B (three stimuli). Therefore, we established a performance criterion for advancing from Stage A to Stage B. Criterion was set as correct performance on the second trials (T2s) on three of the four problems (75%) in a session, in addition to correct performance on T3–T5 of 75% or more in a session, for three sessions in a row.

Stage B. Monkeys that passed criterion (as detailed above) on Stage A commenced Stage B. Testing procedures for Stage B were identical to Stage A with the exception that each problem involved three stimuli, and therefore two of these were unrewarded (and only one rewarded, as per Stage A). Otherwise, testing was structured as in Stage A (i.e. four problems per session, five trials per problem, and consistent reward location for all trials of a given problem).

During the data collection period (May to November 2016), up to two research sessions were conducted per day. All 13 participating monkeys completed at least 10 testing sessions of Stage A (see Table 1), with the minimum number of sessions being 14 and the maximum being 69.

### Statistical analysis

Statistical analysis was done via logit fit GLMMs using the *lme4* package (*Bates et al., 2015*) in R (*R Core Team, 2017*). The following fixed effects were included in at least some of the models: session number (scaled and centred), which was used to determine whether monkeys' performance improved with experience; trial number; the source of information (human, puppet, or individual); and the type of information (whether the information trial was a win or a lose; sum coded). Random effects included in at least some of the models were monkey identity (random intercept effect) and session number and information type by monkey (random slope effects); we used models with 'maximal' random effects structures (*Barr et al., 2013*) in the first instance, removing random slopes followed by random intercepts as necessary to address issues of singular fit or non-convergence. Post hoc analyses were done using the *emmeans* package (*Lenth et al., 2019*).

We used three main response variables: (1) trial *success* indicates whether the correct stimulus was selected on a given trial (binary: 1 = yes; 0 = no); (2) *WSLS* (win-stay, lose-shift) indicates which strategy was used on a particular trial in relation to the information trial (binary: 1 = win-stay or lose-shift; 0 = win-shift or lose-stay); and (3) *repeats* indicates whether the same stimulus was selected on a given trial as was selected in the information trial (binary: 1 = repeated; 0 = not repeated). For Stage A (two stimuli), success and WSLS have the same values. However, for Stage B, these variables may hold different values (i.e. it is possible that, after a lose information trial, a subject can shift and select a second unrewarded stimulus, in which case their success score would be 0 and their WSLS score would be 1).

## Results

Although we had hoped to compare the effectiveness of learning from social demonstrations with that of learning from personal experience, in fact only one monkey reached our performance criterion. Aside from the single individual who reached criterion, there was little evidence that the monkeys were learning anything from their increasing experience with the task. Taking correct stimulus choice (success) on T2 as the measure of interest (i.e. re-selection of stimuli that were rewarded in the information trial, and avoidance of stimuli that were unrewarded), overall performance on T2 of the two-stimulus task (Stage A) was slightly greater than chance, with 56% T2 success (Table 2; observed success: 1,283/2,293 trials; expected: 1,146.5/2,293, $p < 0.001$ in a binomial test). Results of the various models are summarised in Table 3.

### Effect of experience (session number)

A GLMM with a response variable of T2 success, a fixed effect of session number (scaled and centred), and a random effect of monkey identity showed that performance accuracy

**Table 2 Performance summary (all trials, un-averaged) for Experiments 1 and 2.** Statistical tests were two-tailed binomial tests against 50% (chance) performance.

| Study and condition | T2 success after win trials (%) | T2 success after lose trials (%) |
|---|---|---|
| Tablet (Stage A) | | |
|   Social: human | 70* | 50 |
|   Social: puppet | 62* | 47 |
|   Individual | 70* | 40† |
|   Overall | 68* | 44† |
| 3D | | |
|   Social: human | 54 | 55 |
|   Individual | 79* | 31† |
|   Overall | 64* | 46 |

Notes:
* Performance significantly above chance.
† Performance significantly below chance.

**Table 3 Model results summary for Experiments 1 and 2.** Only models that assessed performance as measured by T2 success in Stage A are included in this table. GLMMs included random effects not shown here as well as the fixed effects detailed in the table; please see text for details.

| Analysis | Experiment | Variable(s) included in the model | p-Value and direction of effect |
|---|---|---|---|
| Effect of session number | Experiment 1 (tablet) | Session number | N.S., $p = 0.47$ |
| | Experiment 2 (3D) | Session number | N.S., $p = 0.12$ |
| Effect of trial number | Experiment 1 (tablet) | Trial number | N.S., $p = 0.064$ |
| | Experiment 2 (3D) | Trial number | $p = 0.020$ (later > earlier) |
| Effect of source and information type | Experiment 1 (tablet) | Source | $p = 0.043$ (human > puppet) N.S., $p = 0.61$ (social ndf individual) |
| | | Information type | $p < 0.001$ (win > lose) |
| | | Source * information type | $p = 0.002$ (see Table S2) |
| | Experiment 2 (3D) | Source | N.S., $p = 0.66$ (social ndf individual) |
| | | Information type | $p < 0.001$ (win > lose) |
| | | Source * information type | $p < 0.001$ (see Table S3) |

Note:
N.S., non-significant; ndf, not different from.

did not improve with increasing session number ($b = 0.017$, SE $= 0.024$, $Z = 0.72$, $p = 0.47$). That is, monkeys overall did not improve their task performance with more experience.

### Information type, information source, and interactions

Although the monkeys had not learned the task, it was nonetheless of interest to know how they performed across the different trial types (i.e. individual-win, individual-lose, social-win, and social-lose, with social demonstrations also broken down into human and puppet demonstrations).

 

To explore this, we built a GLMM with T2 success as the response variable; information type, information source, and an interaction of these two variables as fixed effects; a random intercept effect of monkey identity; and session number as a by-monkey random slope effect. We used Helmert contrasts (in the *stats* package in R (*R Core Team, 2017*)) to examine whether performance differed between the three information sources. In Helmert contrasts, the first two levels of a variable are compared to each other, the third level is compared to the mean of the first two levels, and so on (*Fox, 2002*; *Schad et al., 2018*). The key comparisons here are (1) between the two social or vicarious sources of information (human and puppet demonstrations) and (2) between the social and individual sources of information. Therefore, we set the first level as human demonstration, for comparison to the second level, puppet demonstration; and finally, the combined human and puppet levels were compared to the third level, the individual condition. There was a main effect of information type; success after seeing a win trial (68% success) was significantly higher than that after seeing a lose trial (44% success; $b = 0.44$, SE = 0.046, $Z = 9.6$, $p < 0.001$). In addition, success after seeing a puppet demonstration (54%) was significantly lower than that after seeing a human demonstration (60%; $b = -0.13$, SE = 0.063, $Z = -2.0$, $p = 0.043$). There was not a significant difference between the combined social conditions (puppet and human) and the individual condition (55% T2 success; $b = -0.020$, SE = 0.040, $Z = -0.52$, $p = 0.61$). However, there was an interaction between information type and source, indicating a difference between the third level of the source variable (individual condition) and the other two levels ($b = 0.090$, SE = 0.029, $Z = 3.1$, $p = 0.002$). To clarify this interaction, we performed a post hoc analysis using *emmeans*. This indicated that the disadvantage in performance after lose demonstration trials was greater in the individual condition (after a lose, 40% success; after a win, 70% success) than in the two social conditions (after a lose, 49% success; after a win, 66% success; Table S2).

### Repeating previous selections

While T2 success was the main variable of interest, we noted that the pattern of responses possibly indicated a 'stay' bias (i.e. a tendency to repeat the selection from the previous trial, regardless of whether this followed a win or lose information trial). This raised the possibility that the monkeys were influenced only minimally, if at all, by the presence or otherwise of the reward cue during the information trial. That is, they may not have been discriminating between win and lose trials at all. In the individual condition the monkeys may have simply had a tendency to repeat their own previous selections, regardless of any feedback obtained. And in the social condition they could have been subject to fairly non-specific stimulus enhancement effects which drew their attention to previously selected stimuli, again irrespective of whether this had been associated with the reward cue. This would suggest that the monkeys were not using the information trial as 'information' at all, if the outcome of the trial did not influence their choice of stimulus.

Therefore, we analysed another response variable, T2 repeats. This binary variable captured whether, on a given T2, a subject repeated the response (either their own or observed) from the information trial, which was coded as 1, or switched responses, which

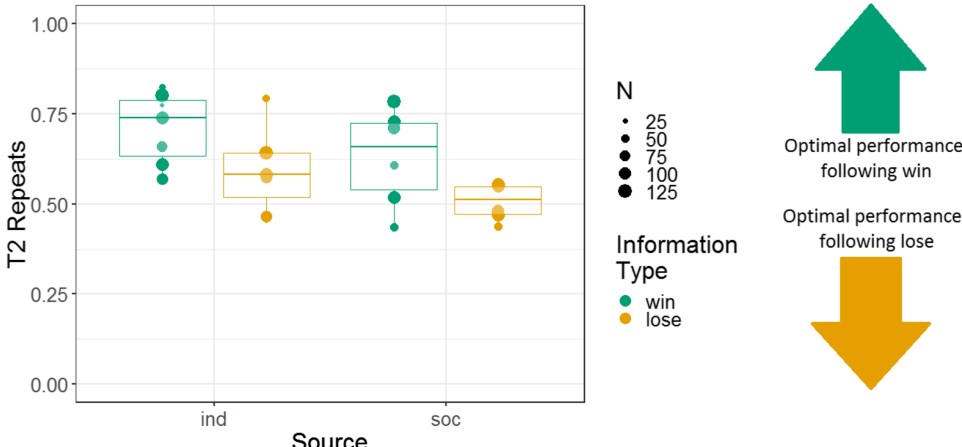

**Figure 3 The effects of source and information type on whether (in T2) squirrel monkeys repeated the selection from the information trial.** Results are from the touchscreen task of Experiment 1. Each point represents the aggregate performance of an individual monkey in the given condition, and the size of each point indicates the number of trials aggregated within that mean. Boxes and whiskers indicate medians and interquartile ranges of performance aggregated by monkey.

was coded as 0. We used a GLMM with T2 repeats as the response variable; information type, information source, and an interaction of these two variables as fixed effects; a random intercept effect of monkey identity; and session number as a by-monkey random slope effect. We again used Helmert contrasts to examine the effect of information source. Human demonstration was the first level, puppet demonstration was the second level, and the individual condition was the third level. There was a main effect of information type; tendency to repeat after a win trial (68% repeats) was higher than tendency to repeat after a lose trial (56% repeats; $b = 0.28$, SE = 0.047, $Z = 6.0$, $p < 0.001$). But note that the proportion of repeats following both information trial types is above 50% (Fig. 3). This indicates a degree of sensitivity to the information in T1, even if the difference is modest. There was no main effect of source, indicating that there were not significant differences in the tendency to repeat between puppet and human demonstrations ($b = -0.065$, SE = 0.063, $Z = -1.0$, $p = 0.31$) or between vicarious and individual selection of a stimulus ($b = 0.11$, SE = 0.065, $Z = 1.6$, $p = 0.10$). The interaction between the information type and the individual vs social conditions was not significant ($b = -0.028$, SE = 0.029, $Z = -0.95$, $p = 0.34$). The interaction between information type and social demonstration type (puppet and human) was non-significant but near the alpha criterion threshold ($b = -0.12$, SE = 0.063, $Z = -1.9$, $p = 0.056$). Further analysis of this trend indicated that the tendency to repeat the puppet's selection (after a win, 62% repeats; after a lose, 53% repeats) was less influenced by information type than the tendency to repeat the human's selection (after a win, 70% repeats; after a lose, 50% repeats).

### Improvement across trials

Because the monkeys appeared not to learn the discrimination rule, and showed a significant tendency to repeat selections from the information trial following both win and

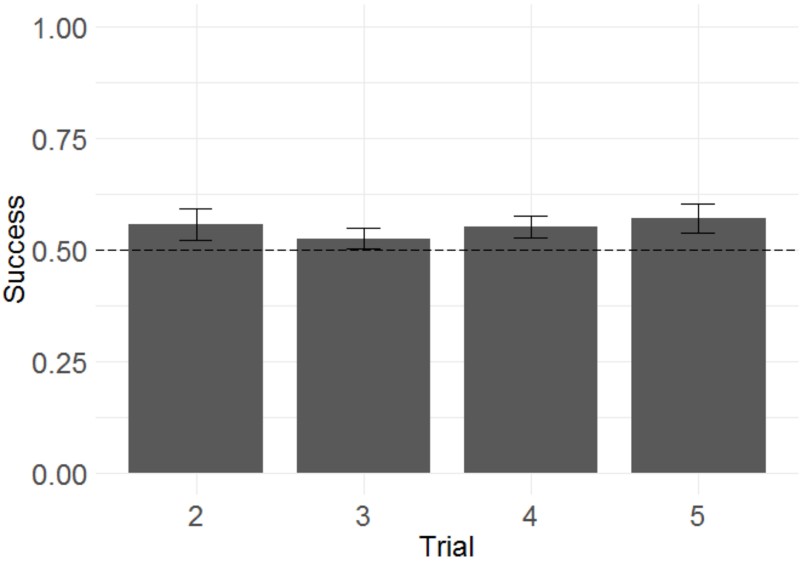

**Figure 4 Success across experimental trials in the touchscreen task of Experiment 1.** Plotted are the means of by-subject means for illustration purposes; statistical tests did not average subject means. The error bars indicate 95% confidence intervals on the means of means. Statistical tests revealed a non-significant trend towards improvement across trials.

lose trials, this raised the question of whether the monkeys were sensitive to task feedback at all. We therefore wished to test whether the additional feedback obtained from the monkeys' selections across all trials for any given problem improved their chance of success (since all information trials were followed by not only the critical test trial T2, but also three further trials using the same problem—T3, T4, and T5—intended to reinforce recognition of the predictive relationship between successive trials using the same pair of stimuli). To determine whether monkeys improved their performance across trials within a problem (as they presumably received confirmatory information about where the rewarded stimulus was or was not), we used a GLMM with success in each trial from T2 to T5 as the response variable; trial number as the fixed effect; a random intercept effect of monkey identity; and session number, information type, and their interaction as by-monkey random slope effects. A non-significant trend towards improvement across trials was demonstrated (Fig. 4; $b = 0.036$, SE $= 0.019$, $Z = 1.9$, $p = 0.064$). It is therefore not possible to conclude that monkeys located the reward more accurately in later than in earlier trials.

### Three-stimulus task (Stage B)

A single monkey (Amarilla) met criterion on the two-stimulus task and was tested on the three-stimulus task (Stage B) for 19 sessions. Amarilla remained in the individual condition throughout the study. Despite having reached criterion on the two-stimulus task, Amarilla failed to reach the same performance criterion (of appropriate stay/shift responses on T2 for 75% of problems in a session, and appropriate stay/shift responses on T3–T5 for 75% or more trials in a session, for three sessions in a row) on the three-stimulus task.

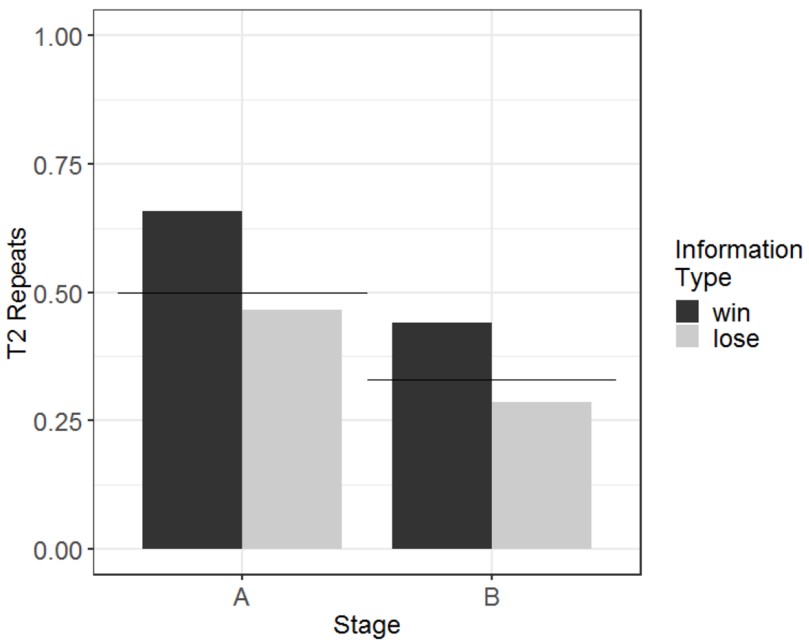

**Figure 5 Probability of repeating in T2 the selection made on the information trial, according to information type, for Stage A and Stage B, for the monkey who passed criterion (Amarilla).** Amarilla was in the individual condition. Expected rates of repetition according to chance are indicated by the horizontal lines (separately for each of the stages).

Our original goal in training the monkeys had been to train them to general competency using this discrimination paradigm, so that we could test the use of social information for more taxing stimulus selection problems. However, this was not possible during the data collection period. Therefore Amarilla's testing sessions were terminated when the planned data collection period ended. Despite this, we did have sufficient data from this subject to perform some analyses on how she transferred her Stage A learning to the novel Stage B context. Her T2 win-stay performance in Stage A was 66% (compared to a chance level of 50%) and in Stage B was 44% (compared to a chance level of 33%). Her T2 lose-shift performance in Stage A was 53% (compared to a chance level of 50%) and in Stage B was 71% (compared to a chance level of 67%). The observed values were compared to chance by use of binomial tests: the only category of trial in which Amarilla's performance differed from chance was Stage A win-stay (observed: 48 win-stays/73 win trials in Stage A; expected: 36.5/73; $p = 0.0095$).

This pattern of performance led to Amarilla finding the rewarded stimulus (and therefore experiencing the reward cue and receiving food reinforcement) in 59% of Stage A T2 trials and 38% of Stage B T2 trials. A summary of the results (showing performance as measured by T2 repeats in order to demonstrate comparisons with chance) is presented graphically in Fig. 5.

To determine whether there was a learning effect (i.e. whether her performance improved with more experience), we used a generalised linear model with T2 WSLS as the dependent variable and session number (scaled and centred) as the fixed effect. This model

showed no effect of session number ($b = 0.12$, SE $= 0.45$, $Z = 0.27$, $p = 0.79$), indicating that Amarilla's performance did not improve with experience in Stage B.

## Conclusion

In general, the squirrel monkeys did not appear to have learned to use the information trial as a cue that ensured success on the test trial(s). Although one subject reached the criterion and transferred from Stage A to Stage B, the success with which she ultimately did so was limited. One potential explanation for the monkeys' limited proficiency with the task was the presentation medium. There is evidence that both young children (*Barr & Hayne, 1999*) and capuchin monkeys (*Anderson, Kuroshima & Fujita, 2017*) learn better when using three-dimensional objects than when viewing videos of a demonstration, a phenomenon termed the video deficit effect. If squirrel monkeys' performance on the touchscreen task was hampered by something like the video deficit effect, we reasoned that performance might be improved by arrays consisting of three-dimensional objects. Therefore, we designed Experiment 2 to reproduce the contingencies of Experiment 1 but to utilise three-dimensional objects as stimuli. Although we intended to reproduce our procedures from Experiment 1 as closely as possible, we decided to use human demonstrations only in the social condition, dispensing with the puppet. This was due to the results of Experiment 1 suggesting that the monkeys performed, if anything, less well following puppet demonstrations, compared with human ones.

# EXPERIMENT 2: THREE-DIMENSIONAL OBJECTS

Experiment 2 was pre-registered at the Open Science Framework (osf.io/phzf2).

## Materials and methods

### Participants

Approval for this project was provided by the Living Links to Human Evolution Research Centre at RZSS Edinburgh Zoo. The same groups of squirrel monkeys participated in Experiment 2 (see Table 1 for biographical details); by the beginning of this study, the number of monkeys had grown via births to 35 individuals. A total of 11 monkeys (seven from the East group and four from the West group) participated in 10 or more experimental sessions (the number specified in the pre-registration as the minimum for their data to be included); a further eight monkeys participated in the experiment but not enough times for their data to be included (<10 sessions each); and 16 monkeys never participated in the study. The data from one subject (Toomi) were excluded, as outlined in the pre-registration, because she developed a side bias (>80% of selections on the same side).

### Ethics statement

Ethical approval was granted by the University of Stirling Animal Welfare and Ethical Review Body under proposal number AWERB (16 17) 117. Animals were not food or water deprived. The research was conducted in line with the guidance provided by *The Association for the Study of Animal Behaviour (2015)* in *Guidelines for the Treatment of*

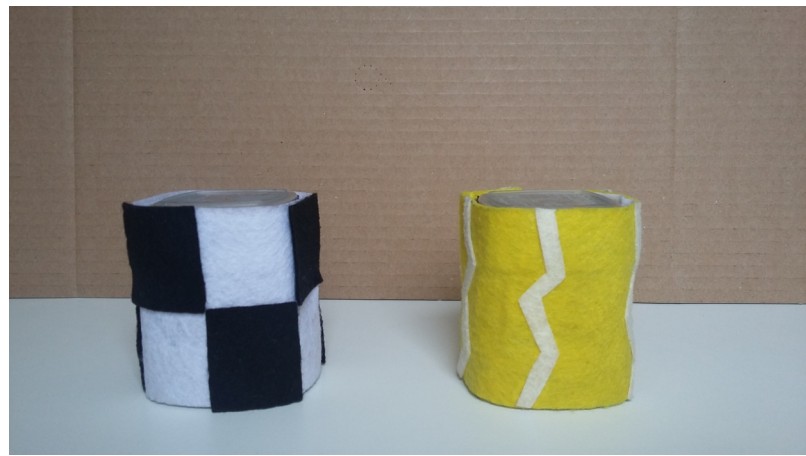

**Figure 6 Two of the cups and felt sleeves in front of the occluder used in Experiment 2.**

*Animals in Behavioural Research and Teaching*, and *The British Psychological Society (2012)* in *Guidelines for Psychologists Working with Animals*.

### Apparatus

Objects were small plastic cups (7 cm tall × 5.5 cm wide × 5.5 cm deep; Sistema, Auckland, New Zealand) covered with differently coloured and patterned felt sleeves (20 cm long × 7 cm tall) that were attached using strips of Velcro (Fig. 6). Objects were presented on top of a mobile trolley. The baiting of the cups was hidden from the monkeys' view with a cardboard occluder. A digital video camera was used to record video of experimental sessions.

### Training procedures

In the course of their previous experience in the research facility, prior to taking part in the experiments reported here, the monkeys had been familiarised with a training activity. In this activity, a single raisin was visibly placed under one of two cups, in trials involving a range of levels of difficulty, and monkeys were then allowed to choose between the cups. Thus, monkeys were already trained to select one of two objects, one of which contained a treat, using their hands. Therefore, no further task training was undertaken before the commencement of Experiment 2. A brief familiarisation session (to allow the monkeys to become familiar with the materials presented to them) involved giving monkeys rewards while the experimenter manipulated the trolley and occluder.

### Testing procedures

Testing procedures were similar to those described above for Experiment 1. Each testing session consisted of four problems, and each problem had five trials. For a single problem, two cups, with different sleeves, were placed, open end down, on the rolling trolley. The order of the use of sleeves was random, so that no particular pattern or colour was systematically rewarded or was systematically paired with any other. To begin a problem, the cardboard occluder was placed in front of the cups; a sunflower seed or raisin reward

was held above the occluder; and the reward was then lowered behind the occluder and placed under a single cup. During the hiding process, both cups were lifted so that auditory cues would not reveal the location of the reward. Then, the occluder was removed and the information trial was carried out, followed by four test trials. Stimuli remained in the same position for an entire problem. The occluder was brought down in front of the cups between trials, so that re-baiting, if necessary, would not be visible to the monkey.

Monkeys remained assigned to the same information source condition they experienced in Experiment 1 (so that if they were in the individual condition in Experiment 1, they remained in the individual condition in Experiment 2). In the individual condition, monkeys selected one of the cups in the information trial, and were rewarded (or not) accordingly. The setup was then re-baited, and monkeys were then allowed to select a cup on the subsequent test trials by reaching through the window toward it or touching or lifting it. In the social condition, monkeys saw a social demonstrator select one of the stimuli in the information trial: this was always the human experimenter, who did not consume the food reward if the rewarded cup was selected but did remove the reward as if it had been consumed. The setup was then re-baited, and monkeys were allowed to select a cup on the subsequent test trials. As in Experiment 1, the purpose of the additional test trials using the same pair of stimuli was to provide scaffolding to allow the monkeys to learn that the reward remained in the same single location for any given pair of stimuli, although T2 again was the critical trial for the purpose of analysis. Monkeys received a food reward on each trial where they selected the rewarded stimulus.

Rewards were randomly allocated to the left or right side for a problem, and patterns of lefts and rights were pseudorandomised so that the reward location would not be predictable and so that rewards would occur equally on each side (to help prevent the development of side biases).

### Statistical analysis

Statistical analysis was done via logit fit GLMMs using the *lme4* package (*Bates et al., 2015*) in R (*R Core Team, 2017*). As in Experiment 1, the main response variables were success, WSLS, and repeats. Fixed effects used in at least some of the models included the session number (centred), trial number, source of information (social or individual; sum coded), and type of information (whether T1 was a win or a lose selection; sum coded). Random effects included monkey identity and side (left or right) of the rewarded stimulus (random intercept effects); similarly to Experiment 1, we removed random effects as necessary to address issues of singular fit or non-convergence.

## Results

As in Experiment 1, overall success on T2 was slightly greater than chance, with 55% T2 success (Table 2; observed success: 305 wins/557 trials; expected: 278.5/557; $p = 0.027$ in a binomial test).

### Effect of experience (session number)

To evaluate the effect of experience, we used a GLMM with T2 success as the response variable, session number (centred) as the only fixed effect, and monkey identity and side of

[1] Due to an oversight, we failed to specify the random effects structure we intended for this model in the pre-registration. This analysis uses the same random effects structure as pre-registered for our main model (see 'Information type and source' below). Alternative random effects structures make no difference to the pattern of results we present here.

the rewarded item as random intercept effects.[1] This model showed no significant change in T2 success in relation to session number ($b = 0.025$, SE = 0.016, $Z = 1.6$, $p = 0.12$); that is, when stimuli were 3D objects, as when they were presented on a touchscreen, monkeys did not improve their task performance with greater experience.

### Information type and source

The full pre-registered GLMM was built, with T2 WSLS as the response variable; information source (social or individual), information type (win or lose), and their interaction as fixed effects; and monkey identity and side of the rewarded item as random intercept effects. This model revealed a main effect of information type (win trial (64% WSLS) > lose trial (46% WSLS); $b = 0.54$, SE = 0.097, $Z = 5.5$, $p < 0.001$) and no main effect of source ($b = 0.051$, SE = 0.12, $Z = 0.44$, $p = 0.66$). There was an interaction between information type and source ($b = 0.55$, SE = 0.098, $Z = 5.7$, $p < 0.001$). To clarify this interaction, we used the *emmeans* package to perform a post hoc analysis. This indicated that in the social condition, there was no difference in performance after seeing win (54% WSLS) and lose (55% WSLS) demonstrations; however, in the individual condition, performance was better after experiencing a win (79% WSLS) than a lose (31% WSLS) in the information trial (see Table S3).

### Repeating previous selections

We analysed the repeats variable for Experiment 2, following the same logic as for Experiment 1 (i.e. to determine whether the monkeys were sensitive to whether they had observed a win or lose trial). Note that analyses using the repeats variable were not pre-registered. We constructed a GLMM with T2 repeats as the response variable; information source, information type, and their interaction as fixed effects; and monkey identity as a random intercept effect. There were main effects of information source and information type, and no interaction between these variables ($b = 0.045$, SE = 0.097, $Z = 0.46$, $p = 0.64$). The tendency to repeat was stronger in the individual (74% repeats) than the social (49% repeats) condition ($b = 0.52$, SE = 0.14, $Z = 3.8$, $p < 0.001$), and the tendency to repeat was stronger after win information trials (64% repeats) than lose information trials (54% repeats; $b = 0.23$, SE = 0.10, $Z = 2.3$, $p = 0.020$) (although note that the tendency to repeat was greater than 50% even following lose trials). Monkeys' performance using the repeats measure is presented in Fig. 7.

### Improvement across trials

[2] Due to an oversight, we failed to specify the random effects structure we intended for this model in the pre-registration. In this analysis, we attempted to use the random effects pre-registered for our main model; however, including monkey identity as a random intercept effect caused the model to have a singular fit. Therefore, only side of the rewarded item was included.

Following the same logic of Experiment 1, we examined whether monkeys improved their performance across trials within a problem, as another means of determining whether monkeys were sensitive to the win/lose feedback they received. We built a GLMM with success in each trial from T2 to T5 as the response variable, trial number as the fixed effect, and side of the rewarded item as a random intercept effect.[2] Trial number did have a significant effect (Fig. 8), indicating that monkeys more frequently selected the rewarded stimulus as trial number increased ($b = 0.092$, SE = 0.039, $Z = 2.3$, $p = 0.020$). The magnitude of the effect was modest, with overall success on T2 of 55% and overall success on T5 of 61%.

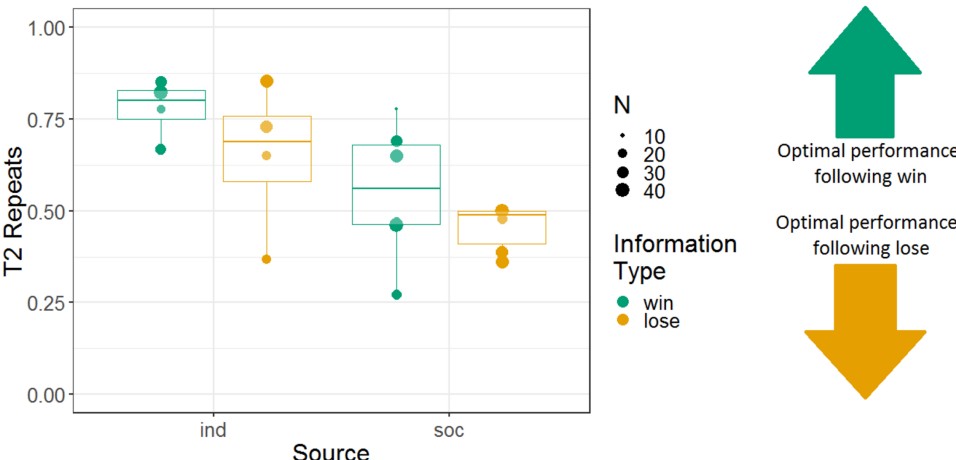

**Figure 7 The effects of source and information type on whether (in T2) squirrel monkeys repeated the selection from the information trial.** Results are from the 3D object task of Experiment 2. Each point represents the aggregate performance of an individual monkey in the given condition, and the size of each point indicates the number of trials aggregated within that mean. Boxes and whiskers indicate medians and interquartile ranges of performance aggregated by monkey.

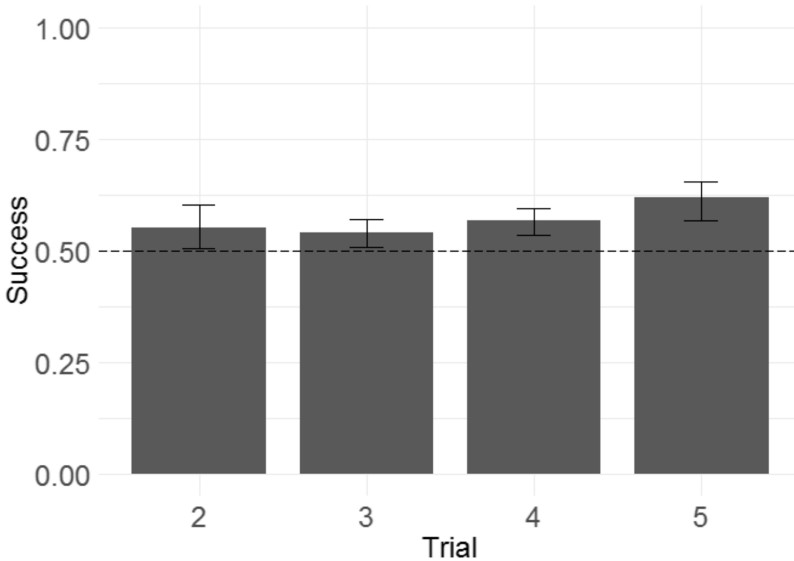

**Figure 8 Success across experimental trials in the 3D object task of Experiment 2.** Plotted are the means of by-subject means for illustration purposes; statistical tests did not average subject means. The error bars indicate 95% confidence intervals on the means of means. Statistical tests revealed a significant effect of trial number.

### Comparison of 2D and 3D presentations

To compare the effect of the presentation mediums used in Experiments 1 and 2, we combined the data sets (using only Stage A data from Experiment 1) and built a GLM with T2 WSLS as the response variable; and information type, information source, presentation medium (touchscreen or cups), and interactions between these variables as fixed effects.[3] Similarly to the models of the data from the two experiments run separately, this GLM

[3] Due to an oversight, we failed to specify the random effects structure we intended for this model in our pre-registration. Although we built a model with a random effect of monkey identity, this model had a singular fit, and side of the rewarded item was not an existing variable in the touchscreen data set. This was the reason for use of a GLM.

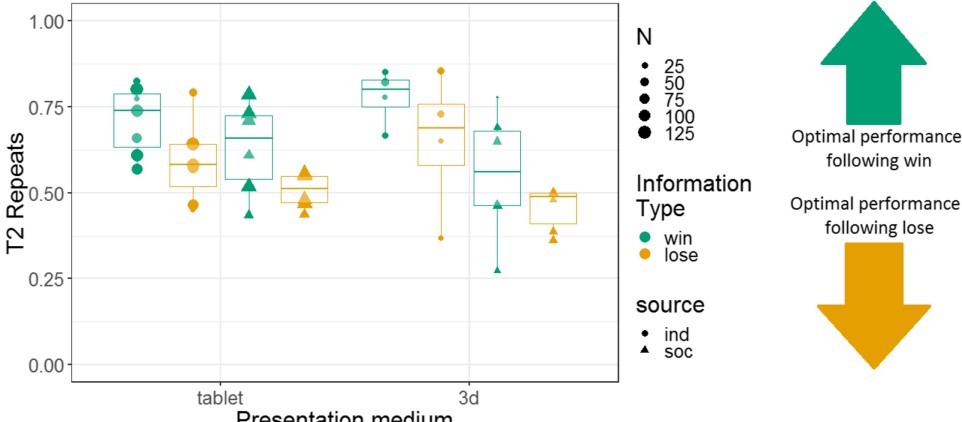

**Figure 9 The effects of presentation medium, source, and information type on whether squirrel monkeys repeated the selection from the information trial on T2.** Each point represents the aggregate performance of an individual monkey, and the size of each point indicates the number of trials aggregated within that mean. Boxes and whiskers indicate medians and interquartile ranges of performance aggregated by monkey.

revealed a main effect of information type (win trial (67% WSLS) > lose trial (45% WSLS); $b = 0.50$, SE $= 0.053$, $Z = 9.6$, $p < 0.001$), while the main effects of the variables information source and presentation medium were non-significant (all $p$s $> 0.67$). There was a significant interaction between information type and source ($b = 0.33$, SE $= 0.053$, $Z = 6.3$, $p < 0.001$), and a three-way interaction between information type, source, and presentation medium ($b = -0.20$, SE $= 0.053$, $Z = -3.8$, $p < 0.001$). For post hoc analysis of this interaction, see Table S4. All other interactions were non-significant.

***Repeating previous selections: a comparison of 2D and 3D presentations***
Additionally, we constructed a non-pre-registered GLMM with T2 repeats as the response variable. This model had fixed effects of information type, information source, presentation medium, and interactions between these variables; and monkey identity as a random intercept effect. This model revealed main effects of information type (more repeats after a win (67% repeats) than a lose (55% repeats) information trial; $b = 0.24$, SE $= 0.053$, $Z = 4.5$, $p < 0.001$) and information source (more repeats in the individual condition (66% repeats) than the social condition (56% repeats); $b = 0.34$, SE $= 0.086$, $Z = 3.9$, $p < 0.001$). There was no main effect of presentation medium ($b = -0.017$, SE $= 0.057$, $Z = -0.29$, $p = 0.77$). However, there was a significant interaction between information source and presentation medium ($b = -0.22$, SE $= 0.057$, $Z = -3.8$, $p < 0.001$; see Table S5). All other interactions in this analysis were non-significant. Results are presented graphically in Fig. 9.

## Discussion
In exposing squirrel monkeys to binary discrimination learning problems, we had originally hoped that they would achieve a performance criterion that could be regarded as indicative of having learned the task contingencies, if given experience of multiple sessions each involving several problems. Establishing task competence would have allowed us to

answer certain questions about the monkeys' responses to information acquired from social observation compared with that acquired from their own personal experience. Although we had limited success fulfilling this original aim, we were nonetheless able to analyse the squirrel monkeys' spontaneous reactions to the task (presumably relatively uncontaminated by the effects of reinforcement learning within the task itself). In addition, we were able to compare the relative effectiveness of two alternative mediums of task presentation—2D touchscreen interaction and 3D object choice—as well as determine whether the monkeys displayed similar patterns of spontaneous responses across these two task types.

Overall, the performance of the squirrel monkeys in both tasks was only very slightly above chance. This contrasts with the performance of pre-school children in a very similar task (*Atkinson et al., 2019*), who spontaneously adopted a win-stay, lose-shift strategy and thus demonstrated highly proficient performance with little or no previous task experience. The squirrel monkeys' performance suggests that even a very simple choice task, involving straightforwardly predictive task cues, may be difficult for them to master unless they have extensive training with the paradigm and/or elevated motivation to succeed (such as calorie restriction, which has been used in other studies carried out by different research groups (*Kangas & Bergman, 2012*)).

In addition, the squirrel monkeys did not show performance improvement across sessions; that is, more experience with the tasks did not lead to better performance. Therefore, inasmuch as they performed marginally above chance, this was not due to learning the task-specific significance of rewarded vs unrewarded information trials, but instead reflected the spontaneous after-effects of having performed an action, or having been exposed to the actions of another, somewhat moderated by whether or not these actions had been associated with rewards and/or reward cues. Although a single monkey did reach our performance criterion on the two-stimulus version of the touchscreen task, her performance upon transfer to the three-stimulus version of the task indicated that, even in this case, there were limits to her understanding of the task contingencies. The monkey's performance suggested that the fidelity of her responses shifted in line with the change in expected response frequencies based on chance dictated by the increase in array size.

Across trials within a problem, squirrel monkeys either did not improve their performance (with the touchscreen, Experiment 1) or improved their performance modestly (with the 3D objects, Experiment 2, from ~55% correct on T2 to ~61% correct on T5). These results may indicate that squirrel monkeys need more than five trials' worth of experience to reliably locate the reward for any given stimulus array. *Harlow (1949)*, for example, gave monkeys between six and 50 trials for each problem; *Darby & Riopelle (1955)* gave macaques six trials per problem; and *Kangas & Bergman (2014)* gave squirrel monkeys 200 trials. However, it should be noted that in those paradigms, unlike the ones presented here, the stimuli changed position between trials. Squirrel monkeys' relatively low problem-level competence in our paradigm would, in turn, make it difficult for them to learn the task-level contingencies.

The presentation medium (2D touchscreen or 3D object) had no main effect on either the monkeys' success on the task or their tendency to repeat a selection. Thus, we found no evidence to suggest that either task was better suited to the abilities of this population, despite the fact that presentation medium did interact with other variables (information type and source).

Squirrel monkeys exhibited a strong tendency to repeat their own selections, following both rewarded and unrewarded information trials. This resulted in unrewarded information trials having success rates below chance in the individual conditions. This tendency to repeat one's own previous selections has been documented by other researchers; notably, macaque monkeys, in an individual learning condition, frequently repeated their own selections, regardless of whether they had been rewarded for that selection or not (*Monfardini et al., 2012*).

However, even given this tendency, squirrel monkeys repeated their own selection more when that selection had resulted in a win than a lose trial. This result indicates some sensitivity to feedback, in that monkeys were more likely to repeat something that had resulted in either a reward or the appearance of secondary reinforcers (auditory click and sunburst image). A similar differential sensitivity to reward (vs non-reward) feedback has been reported for Japanese macaques (*Itoh, Izumi & Kojima, 2001*) as well as baboons and pigeons (*Cook & Fagot, 2009*).

The effect of information source on repetition differed between the two presentation mediums. In the touchscreen task (Experiment 1), the tendency to repeat the selection made in the information trial was not different between the individual and social conditions. Monkeys' repeating behaviour did show an interaction that was near the alpha criterion threshold ($p = 0.056$) between information type and social demonstration type (human or puppet). This trend was in the direction of greater sensitivity to the win/lose distinction for the human's demonstration, compared with that of the puppet. This made it logical to use a human demonstrator only in Experiment 2. While the value of puppet or otherwise artificial demonstrators in other contexts is clear (*Wood, Kendal & Flynn, 2013*; *Krupenye et al., 2016*; *Truskanov & Lotem, 2017*), it was not beneficial for the monkeys' performance in this case.

In contrast, in the 3D object task (Experiment 2), there was a main effect of information source: monkeys tended to repeat their own selections more than those of the social demonstrator. In this Experiment, there was also a main effect of information type on the tendency to repeat, but no interaction between source and information type. This result indicates a sensitivity to source, and to whether a selection was rewarded, but no differential sensitivity to reward information based on the source.

Across both tasks (see combined analyses) the monkeys' tendency to repeat selections was greater in the individual condition than the social condition. In contrast, when children were given a similar task on a touchscreen with information acquired from either a social demonstration or individual exploration, they were largely unaffected by the information source (*Atkinson et al., 2019*). This is likely attributable to the fact that the children apparently understood the predictive value of the information trial (in both conditions), as evidenced by their higher success rates (~73%, c.f. ~55% for the monkeys).

Children's performance appeared to reflect the task-specific relevance of win and lose information trials, whereas (as previously noted) the monkeys' performance presumably reflects pre-existing biases in their behaviour (e.g. stimulus enhancement effects in the social condition, and possibly simple behavioural inertia in the individual condition).

Because the squirrel monkeys did not reach high proficiency levels in our tasks, the results of these experiments cannot be directly compared to results from human children, who demonstrated high levels of task success (*Atkinson et al., 2019*). Such a comparison would have the goal of illuminating the effects of information source across populations, with a view to better understanding the uniqueness of human culture. Drawing comparisons with human performance will require further study with NHP populations that learn to perform the task proficiently. In our ongoing research, we have also tested capuchin monkeys with the touchscreen task, with several individuals so far achieving above-criterion performance (*Kean et al., 2018*).

What our results can reveal, however, is the biases that squirrel monkeys may have when approaching a task. The tendency of squirrel monkeys to repeat selections was not a trained response, and was similar for both mediums of presentation. We speculate that such biases in non-human behaviour could, in certain studies, generate patterns of behaviour that would be consistent with certain interpretations of social learning and/or cultural transmission (i.e. apparent 'copying' of others' behaviour), in spite of the fact that this could be the outcome of mechanisms very different from those driving human social learning and cultural transmission. For example, following observation of an experienced individual, a previously naïve animal may perform the same foraging behaviour. This might occur in the absence of any understanding of either the goal of the demonstrator or even the effect of the behaviour (i.e. the connection with food), although the individual's behaviour would also be consistent with these higher-order cognitive explanations.

In the current studies, our research designs have allowed us to directly compare responses to social demonstrations with those following individual exploration, under equivalent conditions. Furthermore, we were able to evaluate the extent to which the subjects were influenced by the outcome of the response observed or experienced. These comparisons illuminate the fact that the monkeys' slight bias to repeat the behaviour of others was both (a) less strong, if anything, than their tendency to repeat their own behaviour and (b) not motivated by a recognition of the connection between the repeated response and its outcome (since this also occurred—even if to a slightly more limited extent—following unrewarded information trials). Despite our limited success in training the squirrel monkeys on this task, we believe that it is important for researchers interested in animal social learning to pursue designs such as ours, which allow direct comparisons between responses to information acquired from social demonstrations and (equivalent) information acquired from personal experience. Such research may shed valuable light on the mechanisms underlying effects of social influence.

## ACKNOWLEDGEMENTS

We thank the Living Links to Human Evolution Research Centre staff at RZSS Edinburgh Zoo for their training and support throughout this study, and RZSS Edinburgh Zoo for

facilitating this research. We are also grateful to Lindsay Young for her assistance with experimental materials for Experiment 2.

### Funding

This project has received funding from the European Research Council (ERC) under the European Union's Horizon 2020 research and innovation programme under grant agreement number 648841 RATCHETCOG ERC-2014-CoG. There was no additional external funding received for this study. The funders had no role in study design, data collection and analysis, decision to publish, or preparation of the manuscript.

### Grant Disclosures

The following grant information was disclosed by the authors:
European Research Council (ERC) under the European Union's Horizon 2020 research and innovation programme under grant agreement number 648841 RATCHETCOG ERC-2014-CoG.

### Competing Interests

The authors declare that they have no competing interests.

### Author Contributions

- Elizabeth Renner conceived and designed the experiments, performed the experiments, analysed the data, contributed reagents/materials/analysis tools, prepared figures and/or tables, authored or reviewed drafts of the paper, approved the final draft.
- Mark Atkinson conceived and designed the experiments, analysed the data, contributed reagents/materials/analysis tools, authored or reviewed drafts of the paper, approved the final draft.
- Christine A. Caldwell conceived and designed the experiments, authored or reviewed drafts of the paper, approved the final draft.

### Animal Ethics

The following information was supplied relating to ethical approvals (i.e. approving body and any reference numbers):
   The University of Stirling Psychology division and Animal Welfare and Ethical Review Body provided approval for this research. Approval for this research was provided under proposal AWERB (16 17) 117.

### Field Study Permissions

The following information was supplied relating to field study approvals (i.e. approving body and any reference numbers):
   Research was approved by the Living Links to Human Evolution Research Centre at RZSS Edinburgh Zoo.

## Data Availability

The data is available at OSF: Renner, Elizabeth, Mark Atkinson, and Christine A Caldwell. 2019. 'Social Information and Individual Exploration by Squirrel Monkeys in a 3D Stimulus Choice Task.' OSF. June 27. osf.io/tpgej.

## Supplemental Information

Supplemental information for this article can be found online at http://dx.doi.org/10.7717/peerj.7960#supplemental-information.

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
