# Peer review of "Squirrel monkey responses to information from social demonstration and individual exploration using touchscreen and object choice tasks"

_PeerJ, doi:10.7717/peerj.7960_

## Round 0.1 · original submission · Major Revisions

Thank you very much for your submission to PeerJ. I very much appreciate your approach to compare the monkeys' responses to both virtual (touchscreen) and real stimuli. While there has been increasing interest in using touchscreens to test zoo-housed primates' cognition, there is little exploration comparing methodologies directly. Thus, although your experiment did not achieve your original goal, I believe that it represents a useful contribution to the field.

At this time I have been fortunate to receive feedback on your article from three experts in your field. All provided thoughtful and detailed commentaries that will help you to enhance your article as you prepare it for resubmission. I encourage you to respond to each of their comments and describe what edits you made to your article.

In addition to the reviewers' comments, I also had some concerns with your article that I would like you to address. They mostly surround the clarity of your description of your methods and analyses, in particular for experiment 1.

Like the reviewers, I felt that your introduction would benefit from the inclusion of more references to provide context against the background literature. Additionally, and I recognize my own bias here, given that you are running a touchscreen study in a zoo setting and so few currently exist, you might want to comment on why you originally chose such a method and provide context against recent touchscreen studies (or a review thereof).

Regarding your methods, I had trouble understanding fully what you did. I do not think that I could independently replicate them and so I encourage you to clarify your description. Here are some specific points I have regarding experiment 1, but many of these points also relate to experiment 2:

1. For the monkeys that saw a social demonstration, whether the human or the puppet, how was the demonstration performed? What did the model demonstrate? All five trials or just T1? Also, did the demonstrator perform their actions on the monkey’s screen or did the demonstrator have their own screen? How easily could the monkey see the demonstrator’s screen and their actions? Also what was the latency between the demonstrator’s demonstration and the monkey’s T1? How many demonstrations were given (i.e. was a demonstration given before every set of 5 trials?) or just once per day?
2. As written, I am having difficulty understanding your testing paradigm. In T1, was there just one symbol shown on the screen (either rewarded or unrewarded) that the monkey had to select to progress to trials 2-5 (in which two, or three, stimuli were shown on the screen)? If so, please state this clearly in the methods. If not, please clarify what occurred.
3. It appears from your methods that you aimed to run 4 blocks of 5 trials per day with each monkey. If this varied, please provide averages and SDs. How many trials and/or blocks of trials total did each monkey receive?
4. For training (line 158) was there an auditory cue as well, or only a visual cue, as a secondary reinforcer for correct responses? Additionally, how long did training last? Given the increased interest in running touchscreen studies, especially now in zoo settings, this kind of information would be useful to provide.
5. For testing you note “Task stimuli consisted of a variety of shapes (circles, triangles, diamonds, pentagons, and hexagons) of various colours (blue, yellow, black, white, and grey)” – did the stimuli differ across groups of trials? Did the monkeys ever see the same stimuli more than once? If so, I presume it was always either S+ or S- but please confirm this. And were stimuli counterbalanced across subjects? If stimuli were not repeated, it seems highly unlikely that the monkeys would learn the contingency of a stimulus within a single training trial, or even including all five trials. Again, perhaps I am misunderstanding your methods. Apologies if so.
6. You note that, to avoid a side bias developing, you changed the location of the S+ stimulus across conditions but that stimuli were kept in the same location across trials within the blocks of T1-T5. In this case, I would think that the monkeys would just keep pressing the same location where the target stimulus was shown in T1. Was this the case or were monkeys really selecting at random? I would think including “stimuli location” as a factor in your models would be informative to understand their strategies (I believe you did this for Experiment 2, but not for Experiment 1).


Regarding your analyses and results, I also had some questions. Here are some specific points I have regarding experiment 1, but many of these points also relate to experiment 2:

1. Like one of the reviewers, I had trouble accessing your raw data.
2. Why did you set your criteria at 75% to move from A to B? Was this significantly above chance?
3. What random effects did you include in your models? In your results you allude to them, but this should be described in your analysis section. You note a random effect of monkey ID – was this a single metric or did you look at age, sex, or experience with testing as a factor?
4. You note you include trial number to explore learning over time. Was this just 1-5 or was this across all trials presented? Please explain this more clearly.
5. You need to describe your process of model selection. In your results section you describe the results of certain models but do not say why you included the variables you did. Was this due to a priori assumptions or did you methodically compare models to select the one that gave you best fit (e.g., via AIC comparison or LRTs)? Please describe this, otherwise your results appear to be cherry picked.
6. When describing your results you provide some information that would be better presented in your analyses section. Be careful how you delineate these. Or, alternatively, combine them into a single section so you can describe your analyses and results in tandem.
7. Line 270 – note that 56% is not significantly different from chance, as shown by a binomial test so I do not think you should state it is “slightly above chance”
8. On line 285 you state “To explore this, we built a GLMM with T2 success as the response variable”. If I understand your methods, “success” is defined as selecting the S+ stimuli in T2. However, from the monkeys’ perspective might “success” be selecting the stimulus that was shown in T1 (akin to a match-to-sample paradigm)?

Reviewer 1 ·

Basic reporting

The authors aimed to compare squirrel monkey responses to information gained from a social demonstrator versus information gained from individual exploration. Though the monkeys failed to learn the discrimination task presented, both on a touchscreen and with 3D objects, the authors found the monkeys tended to repeat their own selections regardless of whether it was rewarded. In the touchscreen experiment, repetition of selections did not vary between individual and social conditions, while when tested with 3D objects, selection repetition was more likely in the individual condition compared to the social condition. The authors argue that despite failing to train the squirrel monkeys on this task, the study design they employed is unique in providing a means to compare responses based on information source.

The paper is accomplished and well written. The introduction lacks a cohesive argument but has the potential to be bolstered with support from previous studies.

[line 46-50] Explain why the effects of information from a social source are not equivalent to effects of information from personal experience. This seems important to your argument for using this experimental set up, yet what problems exist are not evident here. Also, research exists that explores how individual and social information sources can combat to influence responses. Vale et al. (2017) report how personal experience information measures up when confronted with differing information from social sources in nonhuman primates. Further, I am not sure what claim you refer to as being problematic when based on human behavior or that of our evolutionary relatives.
(Vale, G. L., Davis, S. J., Van De Waal, E., Schapiro, S. J., Lambeth, S. P., & Whiten, A. (2017). Lack of conformity to new local dietary preferences in migrating captive chimpanzees. Animal behaviour, 124, 135-144.)

[line 51-53] This sentence comes too early in the introduction and should appear closer to the end of it. A more thorough argument needs to be constructed before the details of the current study need to be articulated.

[line 59-62] Research supports this, no need to speculate. See below reference.
Horner, V., & Whiten, A. (2005). Causal knowledge and imitation/emulation switching in chimpanzees (Pan troglodytes) and children (Homo sapiens). Animal cognition, 8(3), 164-181.

[line 71-72] It is unclear what questions you are referring to here. Overall, I’d like to see a more coherent argument in this introduction, with more literature to bolster it in order to outline the questions you seek to answer and the specific aims of your study.

[line 101-106] Indeed Monfardini et al. (2012) found information learned socially versus individually lead to different rates of learning in monkeys and humans. However I do not see how the learning process occurred “under significant memory load” as stated in line 103, because Monfardini et al. report on only the first and second responses for each pair of stimuli. Perhaps a bit of clarification on this point is warranted.

Experimental design

As mentioned above, a more organized introduction would include better defining the relevance and more clearly stating the research questions. As for the description of the methods, they are clear and I only have a few notes on them.

[line 160] How was the raisin delivered to the monkey - a dispenser or human?

[line 166] Were research sessions conducted on separate days or could more than one occur within the same day?

[line 170] I recommend including in this first sentence of the testing procedures section that the experimental task consisted of two stages. This way, when first mention of Stage A occurs it is not unexpected.

[line 234–238] The description of Stage B can be written more concisely. Stage A was described well and so it is sufficient to refer to Stage A procedures without having to rewrite them.

[line 261-263] This information belongs in the methods section.

Validity of the findings

I am not well-versed with general linear mixed models and so cannot offer comment on the validity of these statistical analyses.

The content of the results section could benefit from some organization. In general, the results section is for reporting results of the statistical analyses. Interpretation of these results (for example, in lines 263 -267; 559-565; 578-580) is best left for the conclusion/discussion. Also, description of the methods for the statistical analyses pops up throughout this section and could be moved to the Statistical Analysis subsection of the methods.

[line 305 -307] These findings also reported by Monfardini et al., 2012. Reference to this in the discussion would add to the paper.

Additional comments

The article has merit and reports on a valid method for comparing responses in a discrimination task. Overall, the writing can be made a bit more concise by using "that" less frequently and omitting other small filler words. Structurally, I feel some parts of the sections leak into each other, resulting in a cloudy presentation of the research. Stricter adherence to a logical argument in the introduction will greatly improve the paper and support the conclusions described in the discussion.

Reviewer 2 ·

Basic reporting

No comment

Experimental design

There are areas which need some clarification. These are highlighted in the general comments.

Validity of the findings

No comments

Additional comments

Abstract:
Abstract is a good explanation of the experiments, but I think it is missing context and background information. It is clear what you did, but why you did it, and how your results actually inform us is not clear. A sentence or two at the start and end of the abstract should fix this.

Introduction:
Lines 55-69. None of this seems incorrect to state, but it would be appropriate to support these predictions with references.

Exp 1 Methods:
Lines 154. To someone unfamiliar with the training of managed animals, “trained via Shaping” may not make much sense. Elaborate a bit, or include a reference, or both.

Line 160. Was this sunburst reward cue always a black and white circle? (In the example, a correct yellow triangle turns into a sunburst circle) If so, could this not reinforce the selection of circles, or even for the colors black or white?

Line 211. In the puppet trials – Was the experimenter controlling the puppet visible? Otherwise, I do not really see the difference in these two trial types. I think this experimental procedure would have been vastly improved by a conspecific demonstrator (either real, if the experiment allows) or via a video. I think all this addition has done is complicated the analysis and interpretation of the results, without actually adding much. If the authors have a theoretical reason for one being better than the other, I think choosing the the single method would of improved this – as a direct comparison between experimenter and puppet has very little theoretical implications in itself (In my opinion, I suppose!).

Line 211 continued. It is also not clear the animals’ attention to social information was quantified. Did all individuals definitely observe the demonstrator make their selection in all trials? If they were looking away - was this still classed as a social information trial? Or is this information not available. I think quantifying the attention of the animal is extremely important here – and without it, results cannot be interpreted how the authors wish. Please describe the steps taken to make sure animals attended to the social information in the first place. I think this is my biggest concern of the experiment.

Line 228: Where does this criterion come from? It seems a little arbitrary. Justify this criterion level I the text.

Line 234. I think there are bigger differences in Stage A and Stage B (2 stimuli vs. 3 stimuli trials) than the authors claim.

In the unrewarded information trials of Stage B, the first choice made by the monkey may be incorrect (depending on which trial it is). This information does not allow the monkey to infer where the correct choice should be by T2, as there are 2 choices remaining (1 correct, so also, 1 incorrect). This means the monkey could use the information from the first trial perfectly, but also, still be incorrect 50% of the time. This would mean T3 should be Trial where success should be calculated in Stage B, as in this format it takes at least 2 different selections to be 100% sure where the reward it.

Perhaps how this was done will be clearer as you present the results, but at this stage, this isn’t clear.

Edit: It is now clearer in the data, but I think the trial success becomes redundant. If assessing T2 for Stage B, only the WSLS I believe is relevant. If assess T3 for Stage B, then the trial success is relevant. I think the analysis could be simplified to only WSLS data.

Or, since only a single individual progressed to this Stage – I would be tempted to remove this stage completely from the manuscript. As, the data from a single individual can only be very speculative at best, and therefore adds little the manuscript (and if anything, detracts away from Stage A).

Line 240. Mention your random effects here, as I notice in the actual stats they are included. So it would be informative to add this here too.

Exp 1 Results

Line 270. The statistics employed here are not clear. There are no actual results (test and/or P values) reported in the text, or, in the table – simply just an asterix to represent significance. It seems like these are from a binomial test, or something similar, but this isn’t mentioned here or in the methods. Maybe I’ve missed something, but please make this clear.

Exp 2

Much of the comments above are repeated for Exp 2. So please apply these to both experiments.

Discussion

Line 605. I don’t believe motivation is necessarily the issue, as proved by the animals taking part. But yes, I do agree the lack of extensive training is likely the issue, however I understand that it is additionally problematic if the animals are too trained as then you cannot observe a learning strategy as well. As a comment, I think a suitable design would have been to train half of your subjects on the task with only the individual information trial structure, and half of your subjects with just the social information trial structure, and see which group were quicker to ‘master’ the task (e.g. reach your criterion %). I do not suggest any changes or, repeat experiments, just a comment for the future.

Reviewer 3 ·

Basic reporting

The article is well-written in clear English. There are a couple of uses of words which may not be the easiest for particularly non-native speakers to understand and these have been highlighted under minor comments.

I think there are places (noted in minor comments) where the authors could provide some more background literature that would provide more support for their proposals and present the findings more clearly as relevant to many aspects of social learning.

The raw data are made available on OSF. However, I could not open “squirrel_monkey_MDA_LR.csv” and got the error message “Table too large to render”, so the authors might want to check this. The other files opened and appear to contain all the necessary raw data.

The structure of the article is in an acceptable, clear format. The submission is self-contained and presents the results of two linked experiments.

The figures are clear and well described. The box plots are especially useful in displaying the results. The authors might want to put a little bit more space between the y-axis label and the numbering on several of the graphs. It might be good to also add a border to Fig 4 for consistency. All figures appear to be of a good resolution.
There are a lot of results to take in – the figures are very helpful with this, but I wonder if an additional table of results might be useful? It may prove too difficult to amalgamate all the different model results, but if possible it may make them easier to digest.

Experimental design

The article clearly identifies what the aims were with the experimental design. Even though the monkeys did not pass the training criteria and the original aims of the study (of social vs individual information) were therefore impossible to fulfill, the authors make clear what other, interesting findings can be drawn from the results.

The methodology appears to be performed rigorously and to a high standard. The ethical statement could do with being more detailed (i.e. stating that the methods met the appropriate guidelines and that the subjects weren’t food or water deprived etc.). Ethical approval from both the university and research facility are included in supplemental information.

The methods are described in sufficient detail and should be sufficient to replicate. I did not see the R code made available, which would be beneficial to the readers.

Validity of the findings

The study aims to provide new insight into the use of social vs individually acquired information in squirrel monkeys. While the authors were not able to fulfil their original aims since the monkeys to not reach the testing criteria, it still provides information that is of interest and benefit to this field of research, such as a comparison of touchscreen and 3D and an insight into how squirrel monkeys deal with such a task. The conclusions are well stated and linked to the research questions that they identify.The authors identify their speculation as such and make some interesting points.

Additional comments

Summary
This study attempted to examine the effects of source on socially vs individually attained information in an attempt to better understand how humans and NHPs may differ in their use of social information. The monkeys tested did not reach the goal criteria for learning the task and thus the authors were unable to achieve their primary objective. However, the data still provide insights into the monkey’s use of social vs individually learned information and pre-existing biases that are revealed in these tasks.
Overall I found this study to be well written, interesting and of value to the field. It is very useful for other researchers to see “negative results” and interesting that the monkeys did not seem to learn from the task, irrespective of source of information. It is also useful to compare touchscreen vs 3D stimuli as the use of touch screen paradigms is common in this field.

Minor comments:

Abstract – I think you should try and include the nature of the social demonstration (e.g. puppet and human) in the abstract, otherwise the reader might assume as I did from reading it that the social demo was from a conspecific.

L51 – Just a thought, but I’ve heard some discussions lately on abandoning the use of “New World” and “Old World” primates recently because of the Eurocentric origin of the labels. Could consider something like “platyrrhine monkey” instead?
L62 – Might it be relevant to perhaps discuss/cite some of the overimitation literature here? I think in general this paragraph would benefit from some support from the social learning literature.
L95 – You use “vicarious” here and elsewhere – is there another way to more plainly phrase this concept? It might be somewhat confusing for some, especially non-native English speakers.
L104 – Could you perhaps add a bit more detail about the differing effects found?
L144 – Could you add a statement to confirm that the relevant animal ethics guidelines were adhered to?
L291 – Use of vicarious again, perhaps you could use “demonstrated” instead to imply demonstration, but not necessarily by conspecific or human?
Without being an expert in GLMMs, the models seem to be built appropriately and contain the relevant factors as fixed and random effects. The random effects differ across the experiments, but this is explained in the footnotes.
L298 – I’m not familiar with the Helmert contrasts that are used here – I assume that they control for the multiple comparisons between the three information sources (such as when you compare between levels of a factor in a GLMM with “normal” contrasts?) – if so it might be useful to make this clear in the text

Results
Would be good to see the confidence intervals for all the results.
L339 – I think it best not to use phrases such as “approached significance”
L358 – Again, perhaps rather say they improved, but not significantly so, same on L644 & legend of Fig 4
Great that Exp 2 was pre-registered!
L453 – Could you display another measure of effect size here that might make the magnitude easier for the reader to see?

Discussion
L625 – Is there any literature on similar task competence in squirrel monkeys that could be cited here?
L637 – Is there any literature from primates or other animals about positive reinforcement being more effective in learning than negative feedback (in the sense of reward/something being withheld) that could be cited here?
L682 – This is a very interesting point and I’d love to see a bit more discussion on it– can you expand slightly more on how this repetition might be confounded by researchers with social learning, by perhaps giving an example or hypothetical scenario?
Do you think the human not “eating” a reward in Exp1 may have influenced the saliency/success of the social demonstration? Perhaps something worth discussing.

---

## Round 0.2 · Minor Revisions

Thank you for your resubmission to PeerJ. The three reviewers who reviewed your original submission have now reviewed this revised copy. All three agree that you have made considerable improvements to your article, especially in terms of clarity. I agree. However, two of the three reviewers still have a few minor outstanding questions and comments that I ask you respond to before I can accept your article for publication in PeerJ.

Reviewer 1 ·

Basic reporting

The authors have adequately addressed all of my suggestions.

Experimental design

The authors have adequately addressed all of my suggestions.

Validity of the findings

The authors have adequately addressed all of my suggestions.

Reviewer 2 ·

Basic reporting

The manuscript is well written and clear, and now contains sufficient context after the first round of revisions.

Experimental design

No comments

Validity of the findings

Still some minor concerns, addressed below.

Additional comments

I think the authors have made a good attempt to clarify points made by myself, the editor, and the other reviewers, and specifically, the issues I raised have been now clarified. Most of my concerns about have been satisfied except one regarding the success criterion, which I elaborate on more below. On the whole however, I would be comfortable recommending this article for publication, after the authors highlight the following potential issues in their methods.

In response to the following: “To achieve the performance criteria of 75% on T2s and T3-5s in 3 consecutive sessions by sheer chance/random guessing is extremely unlikely (which we calculated in advance). The 75% level, which was above chance but below ceiling, made it (we thought) an appropriate criterion, as it was not too easy nor too difficult, and therefore a useful determinant of whether monkeys grasped the contingency.”

I am still a little unconvinced by this criterion (if I am understanding correctly) in Experiment 1. To advance from Stage A (two-stimulus task), the monkeys needed to achieve 75% on T2 (3 correct choices out of the 4 problems, for 3 sessions in a row). This would mean a monkey who succeeds in 9 out of 12 consecutive T2 trials would reach a threshold to move onto Stage B (Correct?). This criterion is arguably still not strict enough for a couple of reasons. Firstly, this criterion does not represent a statistically significant deviation from chance (traditionally speaking… binomial, p=0.07). Secondly, as this criterion is being constantly tested throughout the 10+ testing sessions, type 1 errors become much more likely as time progesses. After 60 sessions (which at least one individual was exposed too) the animal is very likely to reach this criterion by chance (a multiple comparisons problem), as this criterion would have been tested for 20 times by this point. Ideally, a set amount of sessions should have been completed, and a binomial calculation should have been calculated once at the end of these sessions.

In addition, this criterion for success remains the same for Stage B. This is also problematic because this is a three-stimuli task, and therefore the chance of reaching 75% becomes more difficult. The threshold should have therefore been scaled down for these sessions.

Only a single monkey progressed from Stage A to B in Experiment 1, and because of the above it is not convincing that this was not due to chance. However, setting success criteria this way is seen a lot throughout the literature, so I think it would be harsh to overly-penalise the paper for this approach. However, I do think the authors need to highlight the caution surrounding the possibility for type 2 errors in these methods.

Reviewer 3 ·

Basic reporting

I find the introduction to be much improved with clearer explanations of the problem that the authors are investigating and inclusion of more background literature which provides context for the aims of the present study. Specific terminology has been explained more clearly for the reader.

Experimental design

The methodology used has now been explained in more detail and I believe could be replicated based on this description.

Validity of the findings

R code has been uploaded and problem with raw data resolved.

L356 - the significance of the binomial test and the figure of 56% as above chance seem odd...is it the case that small differences will lead to significant binomial tests when the N are so high (above 2000)?

Additional comments

Abstract: You have added the line, “Both touchscreen and physical object choice tasks appear to be valid methods to study learning in squirrel monkeys, but this population did not master the contingencies in these tasks.” – perhaps it would be better to say that there appeared to be no advantage to one method over the other in the current study

L86 – I would remove “for example”

L92 I would switch “how much” with “to what extent”

In the methods I think you could just make extra clear that T3-T5 were also rewarded (if correct), as the term Information Trial for T1 initially made me question whether the following trials were rewarded (if correct stimuli chosen)

L541 - Did the monkey have to touch the cup to count as a choice of extend his/her hand through the appropriate hole in the plexiglass?

L645 – I assume that this data included only Stage A, but perhaps make this clear.

Overall I find the manuscript to be much clearer in terms of aims, its relevance to our current understanding of social learning research and the detail of its methodology and analyses.

---

## Round 0.3 · accepted · Accept

Thank you very much to responding to the few outstanding review comments. It is my pleasure to accept your article for publication in PeerJ. Congratulations!